# Remote Magnetic Control of Autophagy in Mouse B-Lymphoma Cells with Iron Oxide Nanoparticles

**DOI:** 10.3390/nano9040551

**Published:** 2019-04-04

**Authors:** You-Rong Lin, Chia-Hao Chan, Hui-Ting Lee, Sheng-Jen Cheng, Jia-Wei Yang, Shing-Jyh Chang, Shien-Fong Lin, Guan-Yu Chen

**Affiliations:** 1Institute of Biomedical Engineering, College of Electrical and Computer Engineering, National Chiao Tung University, Hsinchu 30010, Taiwan; yoronglin@gmail.com (Y.-R.L.); chchan0430@gmail.com (C.-H.C.); shengjen@nctu.edu.tw (S.-J.C.); jiawei@nctu.edu.tw (J.-W.Y.); linsf5402@nctu.edu.tw (S.-F.L.); 2Department of Electrical and Computer Engineering, College of Electrical and Computer Engineering, National Chiao Tung University, Hsinchu 30010, Taiwan; 3Gynecologic Oncology Section Department of Obstetrics and Gynecology, Hsinchu MacKay Memorial Hospital, Hsinchu 300, Taiwan; justine3@ms8.hinet.net; 4Department of Biological Science and Technology, National Chiao Tung University, Hsinchu 30010, Taiwan; 5Division of Allergy, Immunology and Rheumatology, MacKay Memorial Hospital, Taipei 10491, Taiwan; htlee1228@gmail.com

**Keywords:** autophagy, magnetic field, iron oxide nanoparticle, proinflammatory cytokine

## Abstract

Autophagy is the spontaneous degradation of intracellular proteins and organelles in response to nutrient deprivation. The phagocytosis of iron oxide nanoparticles (IONPs) results in intracellular degradation that can be exploited for use in cancer treatment. Non-invasive magnetic control has emerged as an important technology, with breakthroughs achieved in areas such as magneto-thermal therapy and drug delivery. This study aimed to regulate autophagy in mouse B-lymphoma cells (A20) through the incorporation of IONPs–quantum dots (QDs). We hypothesized that with the application of an external magnetic field after phagocytosis of IONPs–QDs, autophagy of intracellular IONPs–QDs could be regulated in a non-invasive manner and subsequently modulate the regulation of inflammatory responses. The potential of this approach as a cancer treatment method was explored. The application of IONPs and an external magnetic force enabled the non-invasive regulation of cell autophagy and modulation of the self-regulatory function of cells. The combination of non-invasive magnetic fields and nanotechnology could provide a new approach to cancer treatment.

## 1. Introduction

Autophagy is an important and highly conserved mechanism in the evolution of eukaryotes [1]. In conditions such as nutrient deprivation, hypoxia, pathogenic invasion, or endoplasmic reticulum stress, cells can degrade their intracellular proteins and organelles with lysosomes, thus helping to maintain the balance between the synthesis and degradation of cellular products [2,3].

Many biological applications of autophagy have been identified [4]. For instance, autophagy has been used in combination with standard anti-cancer therapies for anti-tumor applications [5] in several clinical trials to develop novel chemotherapy drugs for lung cancer, multiple myeloma, and lymphoma, as well as breast, colon, and prostate cancers. Some studies have also suggested that autophagy removes damaged organelles and proteins, and could be useful as a tumor-inhibiting mechanism to limit cell growth and genome instability [6,7]. Other studies have indicated that the expression levels of the autophagy-related proteins LC3 and beclin 1 in cancer cells are lower than those in normal epidermal cells. When autophagy was inhibited in mice, cell proliferation was increased, and spontaneous malignancies (i.e., lung cancer, liver cancer, and lymphoma) were induced [8]. In addition, the presentation of associated antigens was decreased, which limited subsequent treatment efficacy. Such data jointly demonstrate that induced autophagy may be useful for anti-tumor applications. Another report [9] provided further compelling evidence that autophagy is related to tumor inhibition. The study showed that the induction of autophagy in cancer cells enhances anti-tumor immunity and results in the reduction of tumor size, thereby confirming the suitability of cancer cells as an experimental model for the study of autophagy in combination with chemotherapy.

Iron oxide nanoparticles (IONPs) have been well explored for their wide range of biomedical applications [10,11,12]. Studies of IONPs in combination with autophagy have demonstrated that IONPs can enter cancer cells through electrostatic attraction and phagocytosis, and subsequently promote cell autophagy and cell death [13,14]. Several research teams have utilized an applied magnetic field and IONPs in cancer treatment, for example by using a magnetic field to generate heat in IONPs for applications in thermal therapy [15]. Other groups have studied the control of proteins inside cells to elucidate mechanisms governing the dynamic architecture of living cells [8,16,17]. However, current knowledge on the application of IONPs and magnetic field control to regulate autophagy in cancer treatment is limited.

In addition to current cancer treatment methods such as conventional chemotherapy and photothermal therapy [18,19,20,21], cancer treatments derived from autophagy are being increasingly explored [22]. This study aimed to regulate autophagy in mouse B-lymphoma cells (A20) through the use of IONPs. With the application of an external magnetic field (20–100 mT) after the phagocytosis of IONPs, their autophagy could be non-invasively regulated. Moreover, the accumulation of IONPs triggered by magnetic force led to the increased level of autophagy and immune response, which subsequently affected the self-regulation in A20 and CT26 cells. However, the increased immune response did not lead to cell death. Therefore, we have suggested a new way for tunable autophagy.

## 2. Materials and Methods

### 2.1. Cell Culture

The mouse B-lymphoma cell line A20 and mouse colon cancer cell line CT26 were used. The culture medium RPMI-1504 (Life Technology-1640; A10491-01; GIBCO, MA, USA.), supplemented with 10% fetal bovine serum (FBS, Biological Industries, Cromwell, CT, USA) and 0.05 mM 2-mercaptoethanol (Sigma-Aldrich, M6250, St. Louis, MO, United States) was applied. A plate culture was established using a Petri dish in a constant-temperature incubator at 37 °C and 5% CO_2_. Passaging was performed when the culture attained 1 × 10^6^–2 × 10^6^ cells/mL. To fix suspended A20 cells for further experimentation, Petri dishes were pre-coated with poly-L-lysine (Sigma-Aldrich, P4707) to facilitate cell adhesion.

### 2.2. Nanoparticle Synthesis

Iron oxide nanoparticles (IONPs) and CdSe/ZnS quantum dots (QDs) were purchased from Ocean NanoTech (San Diego, CA, USA). The particle size distribution of IONPs, QDs, and their composite particles (IONPs–QDs) were analyzed using dynamic light scattering (DLS). The IONPs and QDs used in this experiment had an average particle size of approximately 10 nm. After covalent bonding [23], the composite nanoparticles were re-dissolved to concentrations of 5, 50, and 100 μg/mL in culture medium, and added to a cell-containing well plate for subsequent experiments. Each sample of IONPs–QDs was fixed on a 200-mesh TEM copper grid and dehydrated overnight in an oven. Next, the appearance and size of the materials were observed using transmission electron microscopy (JEOL ARM200F). The surface charges of the materials were analyzed using a zeta potential analyzer, and the magnetization in the materials was analyzed using a superconducting quantum interference device (SQUID, Quantum Design MPMS 5XL SQUID Magnetometer). Samples were measured at 300 K, at a range of −10,000 to +10,000 G, and a sample rate of 500 G/point.

### 2.3. Stimulation of Cancer Cells by Nanoparticles

A20 cells (2 × 10^5^ cells/well) were seeded in a pre-coated poly-L-lysine 12-well plate, and CT26 cells (5 × 10^4^ cells/well) were seeded in a 12-well plate. After incubation for 12 h at 37 °C, the culture media were replaced, and IONPs–QDs were added. After co-culture, subsequent immunostaining and a magnetic field control experiment were performed.

### 2.4. Immunofluorescence

After stimulation by nanoparticles, cells were fixed with 4% formalin at room temperature for 10 min. Formalin was then removed, and the residual solution was washed with phosphate-buffered saline (PBS). Subsequently, 0.05% Tween 20 was added at room temperature and removed, and then the residual solution was washed again with PBS. An LC3 primary antibody (Cell Signaling, #3868) was added for 1 h, and the residual solution was washed twice with PBS. Goat anti-rabbit antibody conjugated with Alexa 488 (Jackson ImmunoResearch Laboratories, Inc., Cambridgeshire, UK) was then added for 1 h, and the residual solution was washed with PBS. Slides were carefully removed for mounting and nuclear staining with 4′,6-diamidino-2-phenylindole (DAPI) and the mounting medium. Phagocytosis of the nanoparticles and activation of the autophagy-related protein LC3 were visualized using a laser scanning confocal microscope, then analyzed using EZ-C1300^®^ software.

### 2.5. Magnetic Field Construction

A magnetic field system used to control magnetic nanoparticles was custom designed. The magnetic field of the system ranged from 20 to 100 mT. The strength of the magnetic field was quantified through measurements using a Gauss meter (KANETEC Co. Ltd., Model: TM-801, Chiyoda-ku Tokyo, Japan).

### 2.6. Multispectral Imaging Flow Cytometry (MIFC)

A20 cells were seeded on a pre-coated poly-L-lysine slide for 24 h and subjected to an applied magnetic field for another 48 h. After PBS washes, cells were then fixed and permeabilized as previously described [24], washed with PBS, and incubated with an LC3 primary antibody (Cell Signaling, #3868, 1:200 dilution) for 1 h at 4 °C in the dark. Cells were then incubated with a goat anti-rabbit antibody conjugated with Alexa 488 for 1 h at 4 °C in the dark. After washing, cells were collected for MIFC (EMD Millipore, Burlington, MA, USA) and analyzed using IDEAS 6.2 (EMD Millipore).

### 2.7. Cell Activity and Cytokine Analyses

After phagocytosis of nanoparticles by the cancer cells and regulation of the cells via a magnetic field, the cells were stained with trypan blue. Cell viability was subsequently calculated using an auto cell counter. The supernatant was stored at −80 °C for further analysis. The supernatant was collected from the magnetic field-exposed cell culture and analyzed using ELISA kits specific for human IL-6.

### 2.8. Statistical Analysis

Data were analyzed using one-way ANOVA and represented as the mean ± standard deviation based on at least three independent capture experiments. Asterisks (*) represent statistically significant differences compared with the control group (* *p* < 0.05 and ** *p* < 0.01).

## 3. Results and Discussion

IONPs conjugated with QDs were synthesized for image tracking in cells. We employed an EDC–NHS method to conjugate IONPs and QDs. The surface of IONPs had a carboxylic acid (–COOH) functional group, and the outer layer was coated with oleic acid and a single polymer layer to enhance biocompatibility. For the QDs, the core material was CdSe/ZnS, and the outer layer was coated with single layers of polymer and PEG(Poly(ethylene glycol) and consisted of an amine (–NH_2_) functional group (Figure 1A). The first step of the experiment was the coupling of QDs with IONPs, which produced red-emitting composite nanoparticles that facilitated subsequent fluorescence and confocal microscopy observations. Through dynamic light scattering (DLS), the particle sizes before and after covalent conjugation were determined, and the average particle size distribution increased from 35.6 ± 6.9 nm to 64.0 ± 29 nm (Table 1). This result showed that QDs successfully bonded to IONPs, and uniformly sized composite particles were obtained. TEM results shown in Figure 1B further demonstrated monodispersity in the size and uniformity of the composite particles. Subsequently, a zeta potential analyzer was used to analyze the surface electrical charge on these materials. Table 1 shows that compared with the original zeta potentials of IONPs and QDs, which were −35 to −15 mV and −20 to +10 mV, respectively, the zeta potential of the composite particles after conjugation changed to +4 to +10 mV. This confirmed that QDs were bound to the IONP surface. Finally, a superconducting quantum interference device (SQUID) was used to analyze the magnetization of composite particles (Figure 1C), and it was confirmed that the particles were still paramagnetic.

Studies of nanomaterials and autophagy have mostly been limited to colon [25] or breast cancer [26,27], and studies on lymphoma are relatively scarce. In this study, the A20 mouse B lymphoma cell line was used as an experimental model. Cellular uptake of IONPs–QDs was examined using confocal microscopy [28]. To explore the efficiency of IONPs–QDs entry into cells, analyses were performed at four different concentrations: 0 µg/mL for the control (untreated) group and 5, 50, and 100 µg/mL for the experimental groups. Image analysis revealed that after the stimulation of A20 cells by IONPs–QDs for 24 h, the amount of material internalized by cells of the 100 μg/mL group was much higher than that of the other groups (Figure 2A). A magnified 3D image showed that nanoparticles were mainly distributed in the cytoplasm (Figure 2B). In addition, a concentration of 100 μg/mL not only allowed a large quantity of IONPs to be delivered into cells, it did not result in significant cell death (Figure 2C). Similar results were observed in CT26 cancer cells (Figure 3). These results are consistent with previous studies and support the notion that magnetic nanoparticles have the potential to safely participate in cell regulation [29,30].

Immunostaining was used to examine whether nanoparticles could induce autophagy in A20 mouse B lymphoma cells [26]. The most important autophagy-related protein, LC3, was analyzed [31]. Usually, LC3 is extensively expressed in the cytoplasm; however, autophagy promoted the accumulation of LC3 and it appeared as puncta within the cell (Figure 4A). In our experiment, only a few green signals (LC3 green puncta) were observed in the control group, whereas for the experimental group, a more pronounced distribution of green puncta was observed in the cytoplasm after 24 h of culturing with nanoparticles (Figure 4B,C). Similar results were observed in CT26 cancer cells (Figure 5). Collectively, these results demonstrate that nanoparticles were delivered to cancer cells and induced autophagy. Compared with the autophagy process induced in cancer cells by other nanomaterials in previous studies [24,32,33], the nanoparticles in our study also induced an aggregation of LC3 into puncta within cancer cells, demonstrating the feasibility of using nanoparticles as autophagy inducers.

A magnetic field was established to enable the adjustment of the magnitude and direction of the magnetic force in real time for subsequent application to cells (Figure 6A). The magnetic field applied to the sample was able to regulate specific cell regions, and with the application of the external magnetic field, phagocytosed nanoparticles in cells aggregated in a specific direction. The possibility of regulating autophagy through this phenomenon was explored. Figure 6A illustrates the cellular response after a 12 h phagocytosis of nanoparticles and 48 h of magnetic field exposure. Then, it shows the relationship between the magnitude of the magnetic field and distance to the sample. As shown in Figure 6B, there were clear movements of IONPs within cells after regulation by the external magnetic field, mostly towards the direction of the magnetic field, demonstrating the feasibility of using the setup to drive IONPs within cells. Subsequently, using confocal microscopy, it was observed that autophagy was modified after the magnetic drive, with more LC3 proteins migrating into the nucleus (Figure 6C). Moreover, magnet-induced accumulation of LC3 puncta was observed in A20 and CT26 cells (Figure 6D and Figure 7). Multispectral imaging flow cytometry was used to quantify the expression of LC3 puncta in cells with or without application of a magnetic field. Figure 6E shows that cells treated with a magnetic field displayed a significantly higher expression level of cells with LC3 than those that were untreated. These data demonstrated the regulation of autophagy in A20 cells by IONPs in a magnetic-field-dependent manner.

This study aimed to achieve control of the non-invasive autophagy regulation of A20 cells, rather than waiting for reactions to spontaneously occur after the injection of nanoparticles. For example, when nanoparticles come into contact with cells, several hours of culturing above a specific concentration is usually required before autophagy occurs in cells and the subsequent cell responses are regulated. However, with application of an external magnetic field, autophagy could potentially be induced in cells within a defined period of time. This level of control could greatly reduce unnecessary damage to other cells during future in vivo treatments, thus increasing the potential application for cancer treatment. In addition, the use of a magnetic force provides clinical advantages compared with optical methods (e.g., deep tissue penetration may not be possible at certain sites) or invasive electrical methods.

There are a few studies on the use of magnetic fields to control IONPs to regulate induced autophagy [32,34]. Previous work demonstrated that the combined use of a magnetic field and IONPs increased the intracellular aggregation of lysosomes and caused cell death [35,36]. Interestingly, in contrast to our previous studies, IONPs and the applied magnetic field did not affect the viability of A20 and CT26 cells (Figure 8A). Our previous studies [37,38] showed that the interaction between nanomaterials (graphene oxide) and cisplatin (CDDP)—a chemotherapy drug—induces LC3 proteins to migrate from the cytoplasm to the nucleus by activating the phosphorylation of imported alpha/beta and histone H1/H4 (e.g., autophagosome completion and autolysosome formation). However, the mechanism enabling IONP interaction with a magnetic field to allow cellular survival is not clear; the size of the particles and magnetic field may be factors that influence the cellular response. Weaker Brownian motion and stronger gravity for particles larger than 100 nm make it difficult to avoid sedimentation [39]. In our previous study, the size of graphene oxide was about 450 nm, which enabled a different function, namely as a drug delivery agent that enhanced the cellular accumulation and retention of CDDP to increase cell death in CT26 cells. However, the size of IONPs–QDs in this study was 52.13 ± 7.0 nm, which was sufficiently small to avoid spontaneous sedimentation under gravity, yet easily accumulate with an external magnetic field, and still possess an enhanced ability for intracellular uptake [16,40]. A magnetic field can also have an impact on signaling pathways and cellular processes [8].

There remains a possibility that our findings are related to a form of autophagy not caused by apoptosis that leads to cell death. One group [27,41] proposed that autophagy can adjust the production of proinflammatory factors and afford protection by resisting the inflammation induced by Dx-SPIONs(dextran-coated iron oxide nanoparticles). Increasing the level of autophagy promoted the differentiation of monocytes and prevented apoptosis. In our study, to explore cytokines after the cellular response to a magnetic field, we measured the level of IL-6, a proinflammatory factor. Figure 8B shows that the level of IL-6 increased in cells with IONPs exposed to a magnetic field. Treating A20 and CT26 cells with IONPs and a non-invasive magnetic field induced increasing levels of proinflammatory cytokines. The mechanism by which a magnetic field combined with IONPs induces an immune response with the retention of cell viability is not clear [42,43]. There are two types of autophagy: autophagy acting to prevent cancer and to promote cancer. Due to the bipolar nature of autophagy in cancer [44,45], the control an external magnetic field to regulate autophagy is a way to overcome these different roles. Our findings provide novel insights into the incorporation of magnetic nanotechnology and magnetic field regulation that could be useful for the development of cancer treatments.

## 4. Conclusions

We successfully applied IONPs and an external magnetic force to non-invasively regulate cell autophagy and modulate the self-regulatory function of cells. A20 cancer cells phagocytosed a significant amount of IONPs, and upon application of an external magnetic field in a specific direction, they aggregated in cells and induced LC3 proteins to migrate from the cytoplasm to the nucleus during autophagy, leading to significant proinflammatory cytokine production. Although we provide no direct evidence relating to the mechanism that triggers this reaction, the application of an external magnetic field to regulate autophagy and inflammation is an innovative concept. Overall, this study demonstrated the feasibility of IONP manipulation with the application of a magnetic field to control autophagy [46]. We believe that our findings will contribute to the development of technology for applying nanotechnology to cancer treatment [22,47], drug delivery [48], and immunology [49,50], and will provide significant research value to relevant fields.

## Figures and Tables

**Figure 1 nanomaterials-09-00551-f001:**
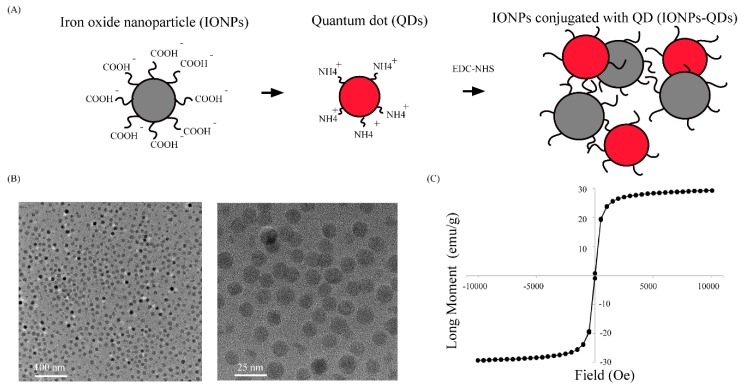
Preparation and characterization of IONPs–QDs. (**A**) Schematic representation of IONPs conjugated with QDs. IONPs (with carboxylic acid groups) conjugated with QDs (with amine groups) using the EDC–NHS method, as a mode of stable tracking in cells. (**B**) Transmission electron microscopy (TEM) image of IONPs–QDs at different scales (100 and 25 nm). (**C**) Magnetization loops of IONPs–QDs.

**Figure 2 nanomaterials-09-00551-f002:**
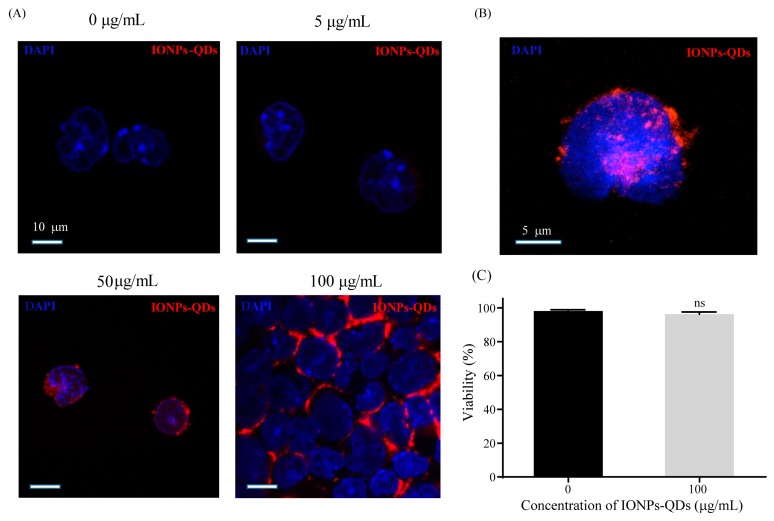
Cellular uptake efficiency of IONPs–QDs and cell viability. (**A**) Cells were incubated with IONPs–QDs for 24 h at 0, 5, 50, and 100 μg/mL. The blue signal (DAPI) represents nuclear staining. The red signal represents IONPs–QDs. Scale bar: 10 μm. (**B**) The magnified 3D image shows that the nanoparticles were mainly distributed in the cell. Scale bar: 5 μm. (**C**) Analysis of cell viability was assessed using trypan blue staining and confirmed using an auto cell counter. ns: not significant.

**Figure 3 nanomaterials-09-00551-f003:**
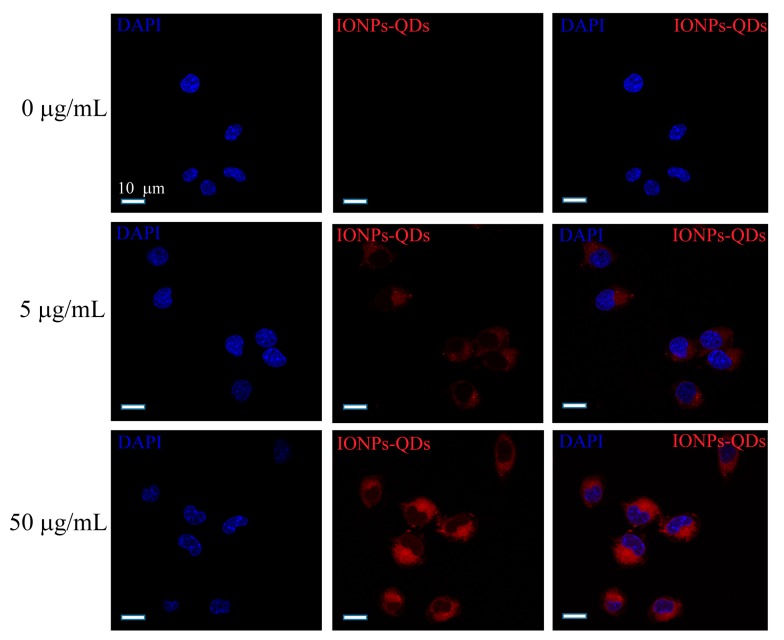
Cellular uptake efficiency of CT26 cells. Cells were incubated with IONPs–QDs for 24 h at 0, 5, and 50 μg/mL. The blue signal represents nuclear staining and the red signal represents IONPs–QDs. Scale bar = 10 μm.

**Figure 4 nanomaterials-09-00551-f004:**
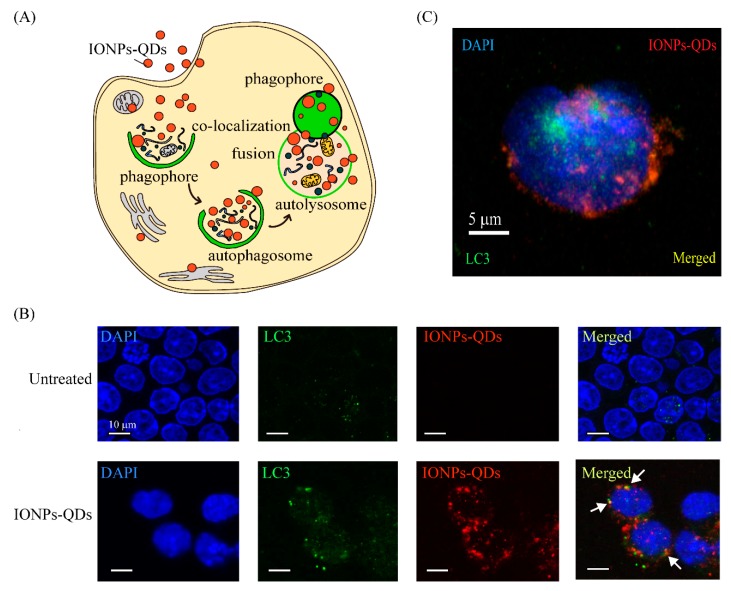
Cellular imaging and IONPs–QDs-induced autophagy. (**A**) Schematic representation of autophagosome co-localization with IONPs–QDs. Procedure of autophagy after cell uptake of IONPs–QDs. (**B**) Co-localization of IONPs–QDs (red) and LC3 marker (green). Scale bar: 5 μm. The white arrow shows co-localization in the cytoplasm (yellow). Scale bar: 10 μm. (**C**) Cellular imaging of a single cell. Cells were stained for the LC3 marker with Alexa-488, followed by UV at 404 nm, and 488 nm laser excitation by confocal microscopy. Scale bar: 5 μm.

**Figure 5 nanomaterials-09-00551-f005:**
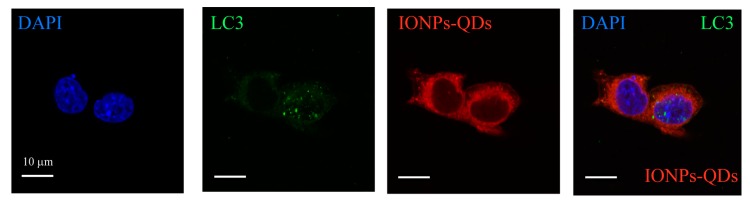
A significant distribution of LC3 puncta (green) was observed in the cytoplasm of CT26 cells. Confocal microscopy image obtained after cells were treated with IONPs–QDs (red) for 24 h; nuclei were stained using DAPI (blue). Scale bar = 10 μm.

**Figure 6 nanomaterials-09-00551-f006:**
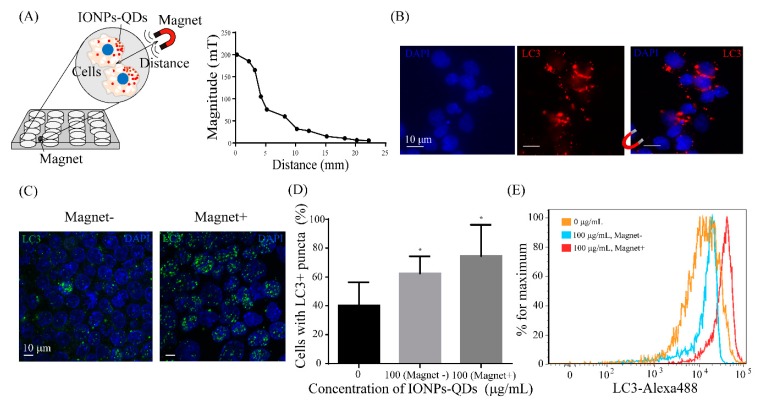
Magnetic field regulation of IONPs–QDs accumulation and autophagy. (**A**) Schematic showing that cell-engulfed IONPs–QDs accumulated based on the applied magnetic field. Cells were seeded on a 12-well plate for 24 h, and then a magnetic field was applied under different conditions. Magnetic field setting and the relationship between the distance and magnetic field magnitude. (**B**) Micrograph showing that the accumulation of IONPs–QDs was modulated by the magnetic field. IONPs–QDs distribution in cells exposed to a magnetic field. Scale bar: 10 μm. (**C**) Cells engulfed IONPs–QDs, resulting in increased accumulation of LC3 puncta (green) and DAPI (blue). Scale bar: 10 μm. (**D**) Quantification of LC3 puncta using confocal microscopy. Quantitative data represent the mean ± S.D. of at least three independent culture experiments. * represents *p* < 0.05 compared with 0 μg/mL IONPs–QDs. (**E**) A20 cells were treated with or without a magnetic field for 48 h and the expression level of LC3 was detected by flow cytometry.

**Figure 7 nanomaterials-09-00551-f007:**
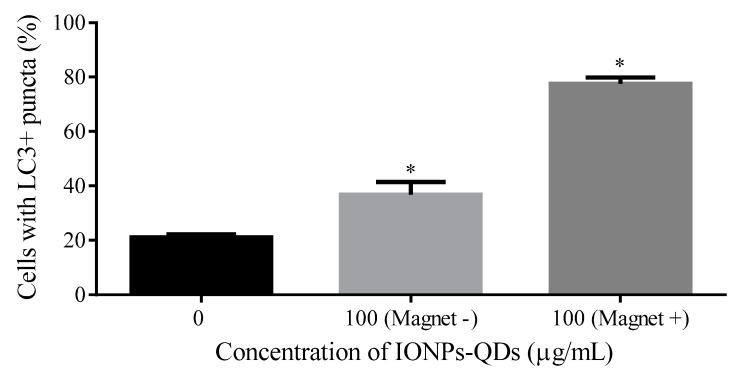
Quantification of LC3 puncta in CT26 cells using confocal microscopy. Three groups: 0, 100 μg/mL without magnetic field (Magnet −) and 100 μg/mL with magnetic field (Magnet +). Note that * represents *p* < 0.05 compared with 0 μg/mL IONPs–QDs.

**Figure 8 nanomaterials-09-00551-f008:**
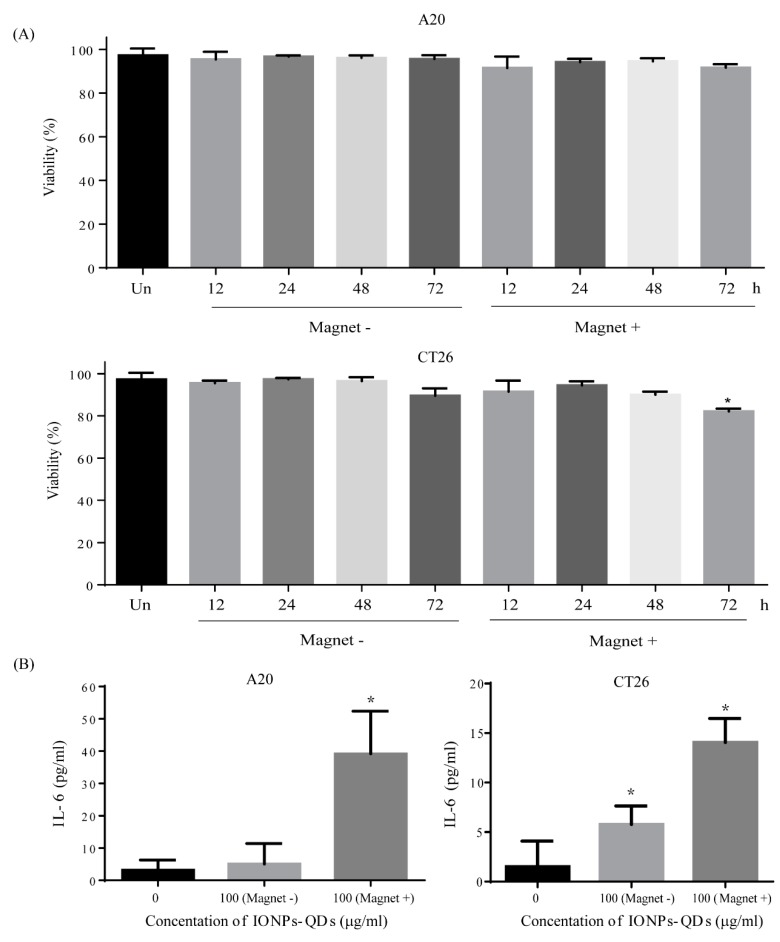
Cell viability and IL-6 level after magnetic field exposure. (**A**) Cells were obtained to verify the effects of IONPs–QDs (100 µg/mL) and magnetic field application at four different time periods: 12, 24, 48, and 72 h. Analysis of A20 and CT26 cell viability using trypan blue dye exclusion and an automated cell counter. (**B**) IL-6 cytokine expression was analyzed at four different conditions: 0 µg/mL IONPs–QDs and 100 µg/mL IONPs–QDs, with or without a magnetic field applied, for 48 h. Quantitative data represent the mean ± S.D. of at least three independent culture experiments. * represents *p* < 0.05.

**Table 1 nanomaterials-09-00551-t001:** Size measurement of DLS and zeta potential of IONPs, QDs, and IONPs–QDs.

Material	Zeta Potential (mV)	Size (nm)
Iron Oxide Nanoparticle (IONPs)	−30 to −50	29.05 ± 6.55
Quantum Dots (QDs)	−20 to +10	20.26 ± 2.23
IONPs–QDs	4.13 ± 10.6	52.13 ± 7.0

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
