# Peer review of "Remote Magnetic Control of Autophagy in Mouse B-Lymphoma Cells with Iron Oxide Nanoparticles"

_nanomaterials, 2019, doi:10.3390/nano9040551_

Reviewer 1 Report

The manuscript by Lin and co-authors describes a method capable to modulate the autophagy in cancer cells.
In particular, they administrated Iron oxide nanoparticles (IONPs) to mouse B-lymphoma cell line A20 and mouse colon cancer cell line CT26. In both cases, the IONPs were internalized by the cells. Authors suggested that the IONPs are included in the autophagosomes and in turn, autophagy can be modulated by using a magnetic field.
The work is interesting, but some aspects must be improved.
Major revisions:
1-Authors have to improve the introduction. The final sentence is too general.
2-Authors have
to improve the results section. They have to move the results from the supplementary file to the main text in order to facilitate the readers. They have to perform more experiments showing the real internalization of the INOPs to the lysosomes. I suggest using other markers to demonstrated the colocalization.
These experiments must be performed also after the application of the magnetic field.

3- The discussion must be improved as well.

Author Response

Point 1. Authors have to improve the introduction. The final sentence is too general.

Authors’ response:

We are very grateful to Reviewer for their time and effort in evaluating our manuscript. According to reviewer’s opinions that “The final sentence is too general”, we have added some discussion in the main text: Moreover, the accumulation of IONPs triggered by magnetic force lead to the increased level of autophagy and immune response, which subsequently affected the self-regulation of in A20 and CT26 cells. However, the increased immune response did not lead to cell death. Therefore, we suggested a new way for tunable autophagy manner.”

Point 2. Authors have to improve the results section. They have to move the results from the supplementary file to the main text in order to facilitate the readers. They have to perform more experiments showing the real internalization of the INOPs to the lysosomes. I suggest using other markers to demonstrated the colocalization.

These experiments must be performed also after the application of the magnetic field.

Authors’ response:

We have already moved all the figures and results from the supplementary file to the main text. We thank for reviewer’s suggestion again and admit that the real internalization of the lysosomes is important. In Zhang et al. work[1], they have to focus on clarifies the mechanism by Iron Oxide Nanoparticles induces autophagosome accumulation and the mechanism of its toxicity. Those results have shown the internalization of the lysosomes on cell organelles and mice organs.

Also, in our previous work [2], we use Graphene oxide (GO) to induce the level of autophagy to explore the underlying signaling mechanism in cancer cells. Colocalization of LC3 and lysosome and increase of LC3-II (both are signs of autophagy flux) were barely observed. Autophagy induction requires autophagosome formation that involves multiple autophagy-related proteins (ATG), Beclin 1, microtubule-associated protein light chain 3 (LC3) and p62. Those related proteins and mechanism are demonstrated in our previous work. Based on the work we have done before, it is shown that GO, another nanomaterial, can induce the toll-like receptors (TLRs) responses and autophagy in cancer cells and confer antitumor effects in mice.

According to these previous data, in this work, we are not focused on other markers to demonstrated the colocalization and mechanism. We aim to regulate the autophagy manner by the non-invasive magnet.

Point 3. The discussion must be improved as well.

We appreciate the reviewer’s comment. We have improved the final paragraph:” There are two types of autophagy, autophagy acting as prevent cancer and promotes cancer. Since the bipolar nature of autophagy in cancer, in that case, thereby control an external magnetic field to regulate autophagy is a way to overcome different role of autophagy. Our findings provide novel insights into the incorporation of magnetic nanotechnology and magnetic field regulation that could be useful for the development of cancer treatments.”

The revisions made in the manuscript have also been highlighted in red, for ease of evaluation.

1. X. Zhang, H. Zhang, X. Liang, J. Zhang, W. Tao, X. Zhu, D. Chang, X. Zeng, G. Liu and L. Mei, "Iron oxide nanoparticles induce autophagosome accumulation through multiple mechanisms: lysosome impairment, mitochondrial damage, and ER stress," Molecular pharmaceutics, vol. 13, no. 7, pp. 2578-2587, 2016.

2. G. Y. Chen, C. L. Chen, H. Y. Tuan, P. X. Yuan, K. C. Li, H. J. Yang and Y. C. Hu, "Graphene oxide triggers toll‐like receptors/autophagy responses in vitro and inhibits tumor growth in vivo," Advanced healthcare materials, vol. 3, no. 9, pp. 1486-1495, 2014.

Reviewer 2 Report

This is a very interesting manuscript which provides novel information about the role of IONPs in the autophagy process. However, some clarification is needed: 

Please give the details about the kind/type of polymeric layer which was used for coated of IONPs to achieve better biocompatibility 

More physicochemical techniques such as DSC or TGA should be provided to better characterized obtained nanoconjugates 

It will be interesting to perform the experiment in the presence of an anticancer drug 

Author Response

Point 1. Please give the details about the kind/type of polymeric layer which was used for coated of IONPs to achieve better biocompatibility 

Authors’ response:

We thank for the Reviewer’s valuable feedback and believe that these suggestions make our manuscript much stronger. Our material, Iron oxide nanoparticles, are coated with monolayer oleic acid, monolayer amphiphilic polymer, and a monolayer of PEG.[website: https://www.oceannanotech.com/products-type/iron-oxide-nanoparticles-5-30nm/functionalized-iron-oxide-nanoparticles/amine-iron-oxide-nanoparticles/amine-iron-oxide-nanoparticles-601.html]

Point 2. More physicochemical techniques such as DSC or TGA should be provided to better characterized obtained nanoconjugates 

Authors’ response:

We thank the reviewer for pointing this out. DSC and TGA are methods of thermal analysis. It can provide valuable physicochemical information about the material. However, to explore the biomedical application of cellular uptake[1, 2], material size is measured by TEM, and hydrodynamic size is measured by DLS have proposed to characterized obtained nanoconjugates. Thus, we didn’t focus on the experiment of DSC and TGA.

Point 3. It will be interesting to perform the experiment in the presence of an anticancer drug

Authors’ response:

We thank the referee for pointing this out and providing us another approach of research. Now, we have already demonstrated the ability of IONPs conjugation with QD as a cellular image agent. In our previous work[3], we use another nanomaterial, Graphene oxide (GO), to induce the level of autophagy and explore the underlying signaling mechanism in cancer cells. It is shown that GO itself can induce the toll-like receptors (TLRs) responses and autophagy in cancer cells and confer antitumor effects in mice. These results have paved the way for iron oxide nanoparticles become a multi-functional platform for image agent, magnet field controllable autophagy response and anticancer drug. Therefore, we didn’t perform the IONPs as an anticancer drug now. However, in the future, we will focus on drug delivery field to explore more interest application.

1. Q. Feng, Y. Liu, J. Huang, K. Chen, J. Huang and K. Xiao, "Uptake, distribution, clearance, and toxicity of iron oxide nanoparticles with different sizes and coatings," Scientific reports, vol. 8, no. 1, pp. 2082, 2018.

2. M. K. Yu, D. Kim, I. H. Lee, J. S. So, Y. Y. Jeong and S. Jon, "Image‐guided prostate cancer therapy using aptamer‐functionalized thermally cross‐linked superparamagnetic iron oxide nanoparticles," Small, vol. 7, no. 15, pp. 2241-2249, 2011.

3. G. Y. Chen, C. L. Chen, H. Y. Tuan, P. X. Yuan, K. C. Li, H. J. Yang and Y. C. Hu, "Graphene oxide triggers toll‐like receptors/autophagy responses in vitro and inhibits tumor growth in vivo," Advanced healthcare materials, vol. 3, no. 9, pp. 1486-1495, 2014.

Round  2

Reviewer 1 Report

no comments